

# Understanding offshore high-ozone events during TRACER-AQ 2021 in Houston: Insights from WRF-CAMx photochemical modeling

Wei Li[1], Yuxuan Wang[1], Xueying Liu[1], Ehsan Soleimanian[1], Travis Griggs[1], James Flynn[1], and Paul Walter[2]

[1]Department of Earth and Atmospheric Sciences, University of Houston, Houston, Texas, USA

[2]Department of Mathematics, St. Edward's University, Austin, TX, USA

*Corresponding author: Yuxuan Wang (ywang246@central.uh.edu)*

**Abstract.** Mechanisms for high offshore ozone ($O_3$) events in the Houston area have not been systematically examined due to limited $O_3$ measurements over water. In this study, we used the datasets collected by three boats deployed in Galveston Bay and the Gulf of Mexico during the Tracking Aerosol Convection Interactions ExpeRiment/Air Quality (TRACER-AQ) field campaign period (September 2021) in combination with the Weather Research and Forecasting (WRF) coupled Comprehensive Air quality Model with Extensions (CAMx) modeling system (WRF-CAMx) to investigate the reasons for high offshore $O_3$. The model can capture the spatiotemporal variability of daytime (10:00-18:00) $O_3$ for the three boats (R > 0.7) but tends to overestimate $O_3$ by ~10 ppb on clean days and underestimate $O_3$ by ~3 ppb during high-$O_3$ events. The process analysis tool in CAMx identifies $O_3$ chemistry as the major process leading to high $O_3$ concentrations. The region-wide increase of long-lived VOCs through advection not only leads to more $O_3$ production under a NOx-limited regime but also fosters VOC-limited $O_3$ formation along western Galveston Bay and the Gulf coast under high-$NO_x$ conditions brought by the northeasterly winds from the Houston Ship Channel. Two case studies illustrate that high offshore $O_3$ events can develop under both large- and meso-scale circulations, indicating both the regional and local emissions need to be stringently controlled. Wind conditions are demonstrated to be important meteorological factors in such events, so they must be well represented in photochemical models to forecast air quality over the urban coastal regions accurately.

## 1. Introduction

The greater Houston area has been designated as ozone ($O_3$) nonattainment by U.S. Environmental Protection Agency (EPA) under the National Ambient Air Quality Standards (NAAQS) standards (Nonattainment Areas for Criteria Pollutants (Green Book), 2023). $O_3$ is a secondary criteria pollutant whose formation is non-linearly dependent on the relative abundance of its precursors: volatile organic compounds and nitrogen oxides. Houston experiences significant anthropogenic emissions of these precursors, mainly from transportation and petrochemical facilities along the Houston Ship Channel (Leuchner and Rappenglück, 2010; Soleimanian et al., 2022). In addition, due to its unique location at the land-water interface, high $O_3$ events in Houston are known to be related to complex



meteorological conditions with the interactions between synoptic and mesoscale circulations. Dry and polluted
continental air masses brought by northerly winds after the cold front passage are often linked with $O_3$ exceedances
(Darby, 2005; Rappenglück et al., 2008; Ngan and Byun, 2011). Extremely high $O_3$ can occur under a land-sea
breezes recirculation, in which the land breeze in the morning transports the pollution-laden air toward Galveston
Bay or the Gulf of Mexico, followed by the return of the aged pollutants in the afternoon by the onshore bay or sea
breeze (Banta et al., 2005; Caicedo et al., 2019; Li et al., 2020). Such high-$O_3$ events in coastal urban regions are
challenging for air quality models to capture as the physical and chemical processes of $O_3$ over both land and water
need to be well-constrained (Caicedo et al., 2019; Bernier et al., 2022).
To understand the interplay among meteorology, emissions, and chemistry, various field campaigns have been
deployed in the Houston area, such as the Texas Air Quality Study in 2000 and 2006 and the Deriving Information
on Surface Conditions from COlumn and VERtically Resolved Observations Relevant to Air Quality (DISCOVER-
AQ) in 2013. A common goal of these field campaigns was to evaluate the predictive ability of numerical weather
and air quality models using the collected observations (Misenis and Zhang, 2010; Yu et al., 2012; Li and
Rappenglück, 2014; Mazzuca et al., 2016; Pan et al., 2017). Although these studies greatly improve our
understanding of the reasons for high ozone events in Houston, they mainly focused on the onshore area due to the
absence of offshore measurements. Higher levels of $O_3$ over water bodies than the adjacent land have been observed
in other coastal regions with poor air quality, such as the Chesapeake Bay and Lake Michigan, due to several factors
including but not limited to the offshore advection of polluted air masses, photochemical productions from local
(e.g., marine traffic) and aged land emissions, shallow marine planetary boundary layers (PBL), the lack of $NO_x$
titration, and low dry deposition rates (Dye et al., 1995; Goldberg et al., 2014; Sullivan et al., 2019; Abdi-Oskouei et
al., 2022; Dreessen et al., 2023). Air quality modeling evaluations against these observations show difficulties in
numerical prediction of $O_3$ over water with an overall positive bias for low $O_3$ and negative bias for high $O_3$ due in
part to the misrepresentation of marine meteorology and PBL (Dreessen et al., 2019; Abdi-Oskouei et al., 2020;
Dacic et al., 2020; Baker et al., 2023). However, to our knowledge, high $O_3$ events off the Houston coast in
Galveston Bay and the Gulf of Mexico have not been systematically examined. The predictive ability of
photochemical models in capturing such events has yet to be quantified.
More recently, the Tracking Aerosol Convection Interactions ExpeRiment/Air Quality (TRACER-AQ) field
campaign revisited the Houston area in September 2021. The campaign implemented a variety of observational
platforms covering both offshore and onshore locations, such as stationary sites, boats, lidar, ozonesondes, and
airborne remote sensing. In particular, instruments onboard three boats continuously collected $O_3$ and
meteorological data from July to October over Galveston Bay and the Gulf of Mexico, which provides a valuable
opportunity to understand the reasons driving high $O_3$ concentrations over water and the $O_3$ non-attainment at air
quality monitors near the Houston coastline. Furthermore, the Texas Commission on Environmental Quality
(TCEQ) has created a new emission inventory for its 2019 state implementation plan (SIP) modeling platform to
conduct photochemical simulations using the Comprehensive Air quality Model with Extensions (CAMx) driven by
the Weather Research and Forecasting (WRF) meteorology. Using the established new emission inventory and



observations, an evaluation of offshore $O_3$ prediction can provide insights into model deficiencies over water and
help improve air quality forecasting in coastal urban regions.
This study aims to improve our understanding of high offshore $O_3$ concentrations in the Houston coastal zone during
the TRACER-AQ 2021 field campaign based on observations and WRF-CAMx modeling, a regulatory model used
by TCEQ. We first evaluate the performance of model simulations of $O_3$ and then investigate the reasons causing
high-$O_3$ events relative to clean days, taking advantage of the process analysis tools from CAMx. Lastly, we present
two case studies to better understand the development of elevated $O_3$ over water. Potential sources of model bias are
also discussed.

## 2. Data and model setup

### 2.1 Meteorological and air quality observations

TCEQ has $O_3$ and other pollutants routinely measured at the continuous ambient monitoring stations (CAMS) across
the Houston region. Some of these stations also observe meteorological variables, such as wind speed and direction,
temperature, and relative humidity (RH). These data can be downloaded from the Texas Air Monitoring Information
System (TAMIS) website. A commercial shrimp boat and a pontoon boat owned by the University of Houston (UH)
were operated mainly on the east and west sides of Galveston Bay, respectively. Another commercial boat, the Red
Eagle, was docked to the north of Galveston Island and typically traveled up to 90 km offshore in the Gulf of
Mexico and occasionally northward through the Ship Channel to the port of Houston. Automated $O_3$ sampling
instruments were installed on the three boats with a compact weather station measuring temperature, pressure, RH,
and wind conditions. The sample inlet was attached to an elevated location on the boats to avoid titration from the
boats' exhausts. Details of these devices can be found in Griggs et al. (submitted). In addition, ozonesondes were
launched from the pontoon and Red Eagle boats on selected days and locations to investigate the vertical $O_3$ profiles.
All the campaign data can be found on the TRACER-AQ website (https://www-air.larc.nasa.gov/cgi-
bin/ArcView/traceraq.2021).
During the offshore operational period of July to October, hourly averaged $O_3$ mixing ratios exceeded 100 ppb
several times. We identified $O_3$ exceedance days when offshore boat $O_3$ observations registered a daily maximum 8-
hour average (MDA8) $O_3$ in exceedance of 70 ppb, the current criteria of the NAAQS for $O_3$. Six episodes with high
$O_3$ were obtained: July 26 – 28, August 25, September 6 – 11, September 17 – 19, September 23 – 26, and October 6
– 9. These episodes are accompanied by at least one CAMS site exceeding the 70 ppb MDA8 $O_3$ threshold,
indicating an extensive land-water air mass interaction.

### 2.2 WRF and CAMx model configuration

This study used the WRF model v3.9.1.1. We set up three domains with different horizontal resolutions that cover
the contiguous United States, Southeast Texas, and the Houston-Galveston-Brazoria region, referred to as domains
d01, d02, and d03, respectively, as shown in Figure 1. The corresponding horizontal resolutions and grid numbers



for d01 – d03 are 12 km × 12 km (373 × 310 grids), 4 km × 4 km (190 × 133 grids), and 1.33 km × 1.33 km (172 ×
184 grids), respectively. All domains have identical vertical resolutions with 50 hybrid sigma-eta vertical levels
spanning from the surface to 10 hPa. Boundary conditions of the two inner domains were generated from the outer
domain.
To select the WRF configurations that best represent the monitoring data, we designed eight model experiments with
different initial and boundary condition (IC/BC) inputs, microphysics options, PBL schemes, data assimilation
method (e.g., observation nudging), and reinitializing techniques. Details of the design and evaluation of each
experiment can be found in Liu et al. (submitted). Based on the campaign-wide evaluation of the modeled
meteorology, the best simulation was used to drive the CAMx model. The model configuration of the best
simulation includes the hourly High-Resolution Rapid Refresh (HRRR) meteorological data as IC/BC inputs, the
local closure Mellor-Yamada-Nakanishi-Niino (MYNN) PBL option (Nakanishi and Niino, 2009), and the Morrison
double moment (2M) microphysics scheme (Morrison et al., 2009) with no nudging and reinitializing techniques
applied. Other settings used for the WRF simulation include the Monin-Obukhov Similarity surface layer (Foken,
2006), the Noah land surface scheme (Chen and Dudhia, 2001), the Rapid Radiative Transfer Model (RRTM)
longwave and shortwave radiation schemes (Iacono et al., 2008), and the New Tiedtke cumulus parameterization
(Zhang et al., 2011).

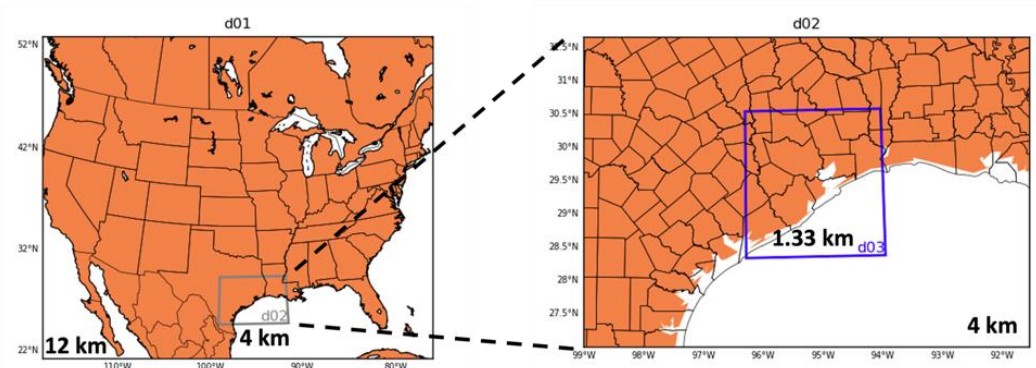


**Figure 1. WRF nested modeling domains and horizontal resolutions.**

This study also used the CAMx model v7.10. The three CAMx domains aligned with the WRF grids but had smaller
spatial coverage. The corresponding horizontal resolutions and grid numbers for domains 1–3 are 12 km × 12 km
(372 × 244 grids), 4 km × 4 km (156 × 126 grids), and 1.33 km × 1.33 km (153 × 162 grids), respectively. All
domains have identical vertical resolutions with 30 vertical levels from the surface to ~100 hPa. The IC/BC inputs
for the outmost domain are from the GEOS-Chem (v12.2.1) global simulation with NEI 2011 nitrogen oxides ($NO_x$)
emissions scaled down to 2021. The Carbon Bond version 6 revision 5 (CB6r5) was used for gas-phase chemistry,



including the inorganic iodine depletion of $O_3$ over oceanic water (Burkholder et al., 2019). The first-order eddy
viscosity (K-theory) diffusion scheme was selected for vertical mixing within the PBL, in which the vertical
diffusion coefficients (Kv) were supplied from WRF outputs. Dry deposition is based on the Wesely scheme
(Wesely, 1989).
Emission files with 12 km and 4 km spatial resolutions from the preliminary 2019 SIP modeling platform provided
by TCEQ are used in the simulation. Since our domains are smaller than those in the SIP modeling, the original
emission files were cropped to match the grid boundaries for CAMx to read properly. In addition, we redistributed
the on-road emissions from 4 km to 1.33 km over the Houston area. The 4 km emission fluxes were first
disaggregated evenly to the 1.33 km grids and then collected onto major roads using a 1 km rasterized road shapefile
to produce on-major-road 1.33 km emissions. Some 1.33 km grid points off the major roads had missing values,
which were filled using a smoothing method that averaged eight nearby grid points. The scaling factors for on- and
off-major-road emissions were kept in order to maintain the on-road emission budget consistent before and after the
spatial redistribution. Finally, total emissions were calculated by adding the 1.33 km on- and off-major-road
emissions. The emissions for other sectors were also similarly interpolated to 1.33 km without separating into no- or
off-major-road temporary emissions. The redistributed emissions were tested to perform better in capturing the on-
road distributions than using the Flexi-nesting function in CAMx (Figure S1), which can regrid the emissions on the
fly.
The simulation was performed for two periods, July 20 – 30 and August 20 – October 13, to cover the six high-$O_3$
episodes defined in Section 2.1. A 10-day spin-up before each period was applied. Other days in the two periods are
considered clean scenarios with low $O_3$ concentrations. Process analysis, including integrated process rate analysis
(IPR), integrated reaction rate analysis (IRR), and chemical process analysis (CPA), was turned on when running the
model. IPR contains $O_3$ change rate from several chemical and physical processes, such as chemistry (CHEM),
horizontal and vertical advection (ADV) and diffusion (DIF), and deposition (DEP). IRR provides detailed
information about the reaction rate of all the chemical reactions in the CB6r5 scheme. CPA improves upon IRR by
computing parameters useful for understanding $O_3$ chemistry, such as $O_3$ production rate and regime. The $O_3$
formation regime is determined based on the ratio of hydrogen peroxide ($H_2O_2$) production rate from hydroperoxyl
radical ($HO_2$) self-reaction to nitric acid ($HNO_3$) production rate from hydroxyl radical (OH) reaction with nitrogen
dioxide ($NO_2$), in which $P(H_2O_2)/P(HNO_3) < 0.35$ indicates a VOC-limited regime and $\geq 0.35$ indicates a NOx-
limited regime (Sillman, 1995). There is no transition scheme available in this method.
**3. Results**
**3.1 Evaluation of $O_3$ simulations**
The time series of the daytime (10:00 – 18:00) mean $O_3$ at the three boats are shown in Figure 2a, and the evaluation
statistics are listed in Table 1. The evaluation excludes nighttime data to reduce the effects from land as the boats
stayed at the dock at night. Indeed, an hourly time series evaluation with nighttime data included (Figure S2 and



Table S1) shows a larger bias between modeled ozone and boat observations. The spatiotemporal variability of
daytime $O_3$ at the three boats is well captured by the model with a correlation coefficient (R) value greater than 0.70.
Overall, the model overestimates daytime $O_3$ by 4.57 ppb (11%), 7.82 ppb (22%), and 4.35 ppb (9%) for the
pontoon boat, Red Eagle, and shrimp boat, respectively. On episode days, high $O_3$ mixing ratios can be found over
Galveston Bay and the Gulf of Mexico (Figure 2b). The model captures some of the variability (R=0.42 – 0.51),
with negative mean bias (MB) values of ~4.5 ppb (8%) for the pontoon and shrimp boats and a nearly unbiased
simulation (MB=0.05 ppb) for the Red Eagle boat. While the $O_3$ variability is better predicted on clean days (R=0.69
– 0.76), the model shows higher values of MB than those on high-$O_3$ days ranging from 9.15 ppb (29%) to 11.28
ppb (41%), which drives the overall model overestimation.

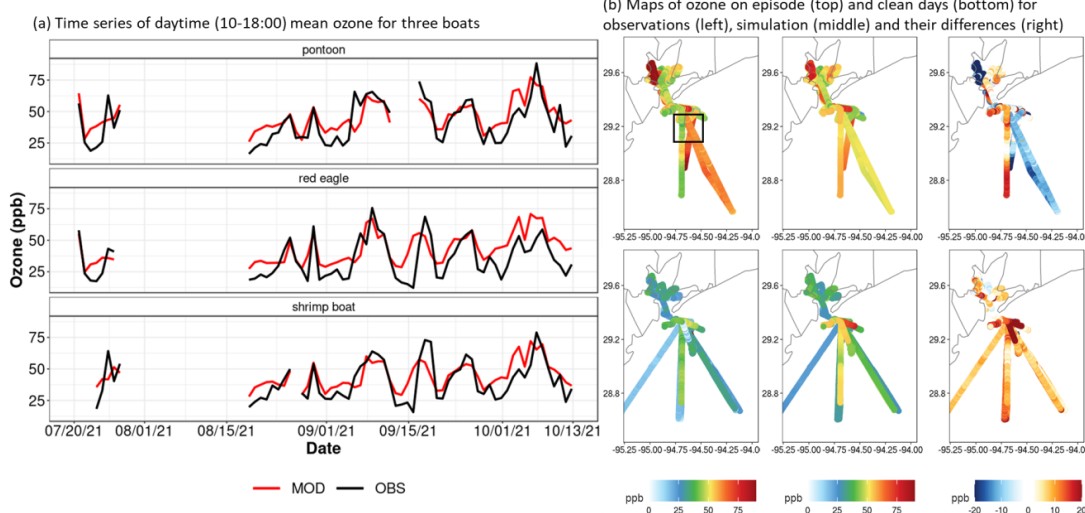


**Figure 2. (a) Time series of daytime (10:00 – 18:00) mean ozone for observations at three boats (black) and simulations**
**(red). (b) Maps of observed (left column), simulated (middle column), and their difference (right column) of ozone during**
**ozone episodes (top row) and clean days (bottom row). The black box shows the selected offshore region for process analysis**
**in the next section.**

While we did not find any previous efforts modeling offshore $O_3$ in the Houston area to compare our results, an
evaluation against onshore measurements can help validate our model performance. The time series of the daytime
mean $O_3$ from simulations and observations from CAMS sites are displayed in Figure 3, and the evaluation statistics
are summarized in Table 2. The model captures the onshore $O_3$ variability (R=0.79) with an overall overestimation
of 7.89 ppb (20%), mainly due to the high positive bias of 10.93 ppb (34%) on clean days. This result is comparable
with the model performance from previous studies focusing on the same area (e.g., Xiao et al., 2010; Pan et al.,
2015; Kommalapati et al., 2016), which further verifies the reliability of our model settings.



**Table 1. Daytime (10:00 – 18:00) ozone evaluation metrics at three boats, including the observed and simulated mean values, correlation coefficient (R), mean bias (MB), normalized mean bias (MB), mean absolute error (MAE), and root mean square error (RMSE).**

| Boat | Period | Observed mean (ppb) | Simulated mean (ppb) | R | MB (ppb) | NMB (%) | MAE (ppb) | RMSE (ppb) |
|---|---|---|---|---|---|---|---|---|
| pontoon | all days | 41.18 | 45.76 | 0.77 | 4.57 | 11.12 | 9.75 | 11.57 |
| | ozone episode | 58.57 | 54.21 | 0.51 | -4.36 | -7.44 | 8.34 | 11.31 |
| | clean days | 32.06 | 41.33 | 0.76 | 9.27 | 28.93 | 10.50 | 11.71 |
| Red Eagle | all days | 34.86 | 42.69 | 0.71 | 7.82 | 22.45 | 11.15 | 13.42 |
| | ozone episode | 51.20 | 51.25 | 0.42 | 0.05 | 0.08 | 9.71 | 11.92 |
| | clean days | 27.60 | 38.88 | 0.69 | 11.28 | 40.89 | 11.80 | 14.03 |
| shrimp boat | all days | 39.99 | 44.35 | 0.73 | 4.35 | 10.89 | 9.15 | 11.47 |
| | ozone episode | 57.22 | 52.22 | 0.43 | -5.00 | -8.74 | 8.88 | 11.65 |
| | clean days | 31.17 | 40.32 | 0.69 | 9.15 | 29.36 | 9.28 | 11.38 |

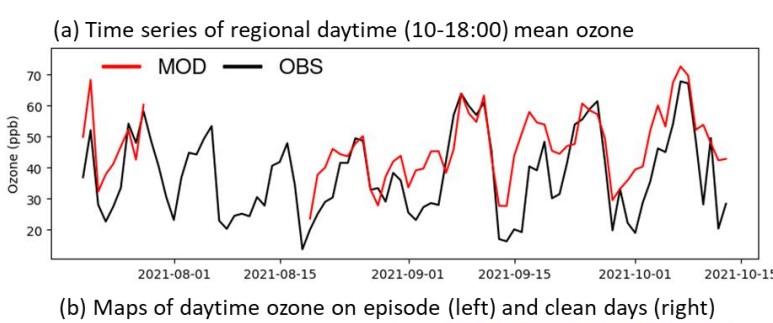

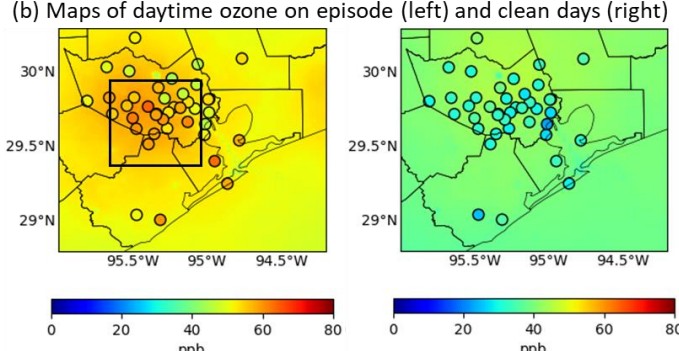

**Figure 3. (a) Time series of daytime (10:00 – 18:00) mean ozone for observations at CAMS sites (OBS; black line) and simulations (MOD; red line). (b) Maps of observed (points) and simulated (background) daytime ozone during ozone episodes (left) and clean days. The black box shows the selected onshore region for process analysis in the next section.**




**Table 2. Daytime (10:00 – 18:00) ozone evaluation metrics at CAMS sites. The metrics are the same as in Table 1.**

| Sites | Period | Observed mean (ppb) | Simulated mean (ppb) | R | MB (ppb) | NMB (%) | MAE (ppb) | RMSE (ppb) |
|---|---|---|---|---|---|---|---|---|
| CAMS | all days | 38.87 | 46.76 | 0.79 | 7.89 | 20.32 | 9.41 | 11.72 |
| | ozone episode | 54.63 | 56.17 | 0.64 | 1.54 | 2.81 | 5.31 | 7.15 |
| | clean days | 31.34 | 42.28 | 0.64 | 10.93 | 34.88 | 11.35 | 13.37 |



**Figure 4. Ozone vertical distribution from the afternoon (12:00-18:00) ozonesonde launches (Obs; black lines) and**
**simulations (Mod; red lines) at Galveston Bay averaged on clean days (dashed lines) and ozone-episode days (solid lines).**
**The Gulf of Mexico only sampled ozone on high-ozone days.**
We also evaluated the modeled vertical $O_3$ profiles against the afternoon (12:00-18:00) ozonesondes launched over
Galveston Bay and the Gulf of Mexico. During the study period, there were five and nine afternoon launches over
Galveston Bay on clean and $O_3$-episode days, respectively, while the Gulf of Mexico only had five afternoon
launches during high-$O_3$ events. The average $O_3$ profiles from these launches are shown in Figure 4. Free
tropospheric $O_3$ with altitudes greater than 2 km is underestimated for both locations on both clean and $O_3$-episode
days, which indicates the long-range transported $O_3$ is underrepresented by the model. Over Galveston Bay, the
overestimation of $O_3$ within the mixed layer below 2 km on clean days changes to underestimation on episode days,
and the underestimation increases from 5 ppb at the surface to 10 ppb near 1 km. This underestimation of $O_3$ in the
mixed layer on episode days can be partly explained by the missing high-$O_3$ layer between 2 – 3 km, which can be
mixed down when the cap inversion is weak (Liu et al., submitted). There is an approximately 10 ppb
underestimation across all altitudes below 4 km over the Gulf of Mexico. An ozonesonde from the Gulf of Mexico
on September 9 recorded high ozone up to the top of the marine layer at 370 m, which is missed by the model and
leads to the highest bias. This case will be discussed in the case study of Section 3.3.





To conclude, despite the overall model bias for vertical $O_3$ distributions, the acceptable model performance for
offshore and onshore $O_3$ prediction at the surface indicates that the modeling system can be applied to conduct
process analysis and help identify the processes influencing high $O_3$ concentrations over the water surface.
**3.2 Process analysis over the Gulf of Mexico**

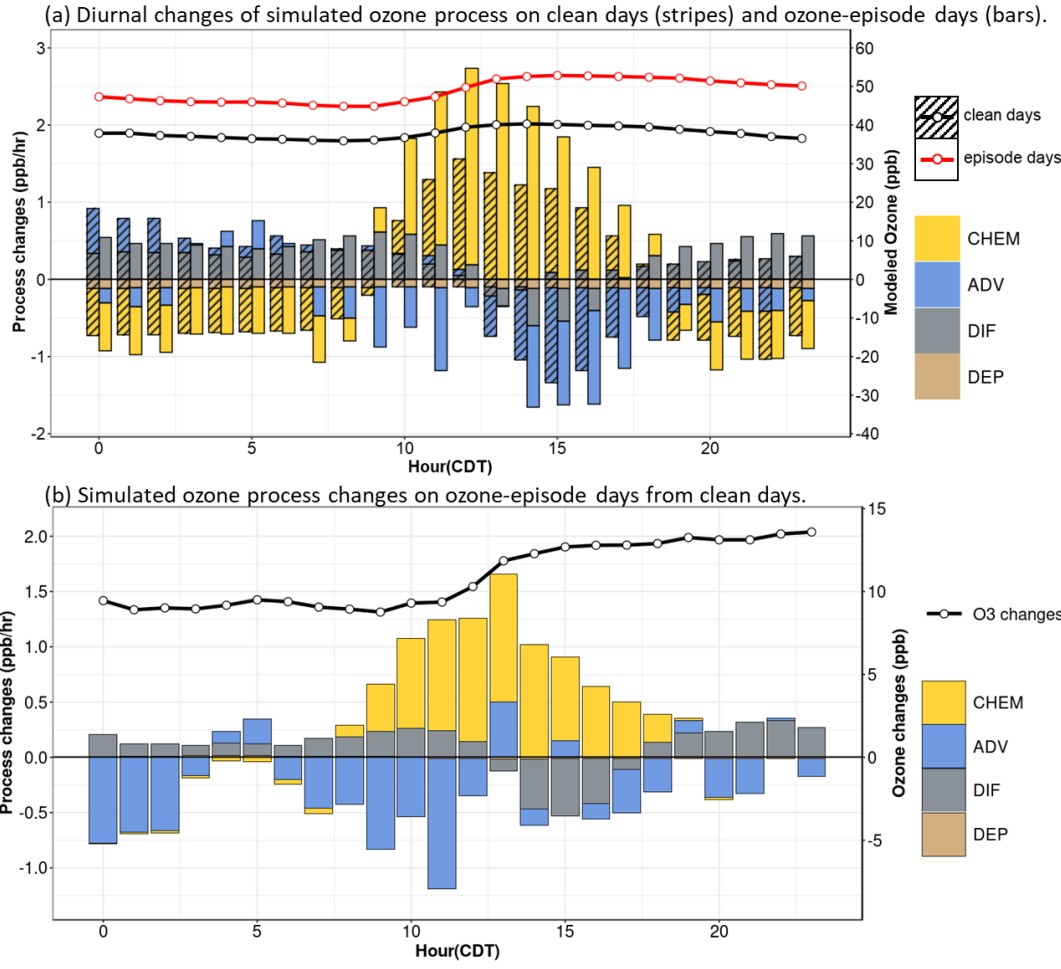


**Figure 5. (a) Diurnal changes of simulated ozone processes over the Gulf of Mexico (black box in Figure 2), including chemistry (CHEM), advection (ADV), vertical diffusion (DIF), and deposition (DEP) on clean days (stripes) and $O_3$-episode days (bars) integrated across the lowest five model layers. Overlaid lines and points are simulated hourly ozone on clean (black) and $O_3$-episode (red) days. (b) Process (filled bars) and $O_3$ (black line) changes during high-$O_3$ episodes relative to clean days.**

This section examines how the CAMx simulated $O_3$ processes change during high-$O_3$ episodes relative to clean
days. The process analysis is calculated over a subregion of the Gulf of Mexico with high $O_3$ mixing ratios observed
(black box in Figure 2b) and integrated across the lowest five model layers comparable to the morning PBL heights





over water. The diurnal average of each process on clean and O$_3$ episode days is shown in Figure 5a. Chemistry
(CHEM) is the major O$_3$ source during daytime and becomes the primary O$_3$ sink after sunset. Advection (ADV)
serves as a pathway for an O$_3$ sink for most hours, especially during the day, while vertical diffusion (DIF) mostly
contributes as an O$_3$ source. Deposition (DEP) constantly removes O$_3$ from the atmosphere at all hours, yet with a
marginal value of 0.1 ppb/hr. Similar patterns can be found over the Houston urban area with a much bigger
magnitude (Figure S3). During high-O$_3$ events, CHEM is the most important process causing higher O$_3$ levels over
water relative to clean days, followed by vertical DIF (Figure 5b). We examined the simulated O$_3$ vertical profiles
and PBL heights averaged over the process analysis region on clean and episode days in Figure S4. O$_3$ across the
entire profile is higher on episode days than clean days, indicating an elevated O$_3$ background on high-O$_3$ days. In
addition, the O$_3$ gradient above and below the PBL is also higher on episode days, especially during morning hours,
which can induce more vertical diffusion if downmixing occurs from above the PBL when the capping inversion is
weak (Liu et al., submitted).
The CPA analysis can provide more insights into the enhanced O$_3$ production during high-O$_3$ events. We first
investigated the rates of HO$_2$ self-reaction and OH reaction with NO$_2$ in Figure 6a-b since they are used by the
model to determine the O$_3$ chemical regime. A region-wide increase in the HO$_2$ self-reaction rate leads to the
enhancement of PO$_3$ under a NO$_x$-limited regime (Figure 6c). Similarly, the frequency of PO$_3$ under a NOx-limited
regime also increases regionally (Figure S5). The frequency at each grid cell is the ratio of the number of hours with
a greater than zero NO$_x$-limited PO$_3$ to the total midday hours (11:00 – 15:00) during the study period. HO$_2$ is
formed following the oxidation of VOCs by OH. Thus, we further compared the OH reactivity of VOCs averaged
from 11:00 to 15:00 on clean and episode days in Figure 7. Isoprene has the highest contribution to the total VOC
reactivity on the land, but its reactivity does not increase during high-O$_3$ events. Instead, paraffin, formaldehyde, and
acetaldehyde are the three VOCs experiencing the highest increase of reaction rate with OH over both land by 0.22
ppb/hr (84%), 0.19 ppb/hr (45%) and 0.15 ppb/hr (73%) and water by 0.18 ppb/hr (114%), 0.15 ppb/hr (44%) and
0.11 ppb/hr (82%), respectively, which indicates a higher contribution from regional transport on episode days as
they are relatively long-lived VOCs capable of traveling long distances. Indeed, the paraffin IPR analysis shows that
the ADV process dominates the increase of paraffin during morning hours from 06:00 to 11:00 over water (Figure
S6). The trajectory analysis focusing on two O$_3$ episodes in September shows air masses were transported from the
northern/central states (Soleimanian et al., submitted), consistent with the wind directions demonstrated in Figure 6.
Such wind conditions can also bring NO$_x$ emissions from the Houston Ship Channel downwind towards the western
side of Galveston Bay and the Gulf of Mexico, causing a higher OH reaction rate with NO$_2$ (Figure 6b) and
enhanced PO$_3$ under a VOC-limited regime (Figure 6d) therein.

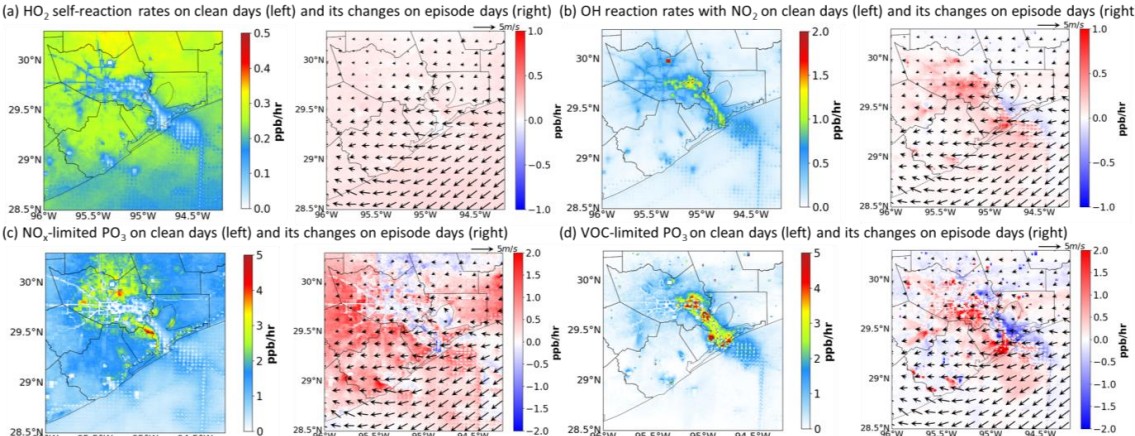

**Figure 6. Maps of the rate (ppb/hr) of HO₂ self-reaction (a), OH reaction with NO₂ (b), ozone production (PO₃) under NOₓ-limited (c) and VOC-limited (d) regimes on clean days (left) and its changes under episode days (right) during midday (11:00 – 15:00). Black arrows indicate the simulated wind speed and directions averaged on high-O₃ days.**

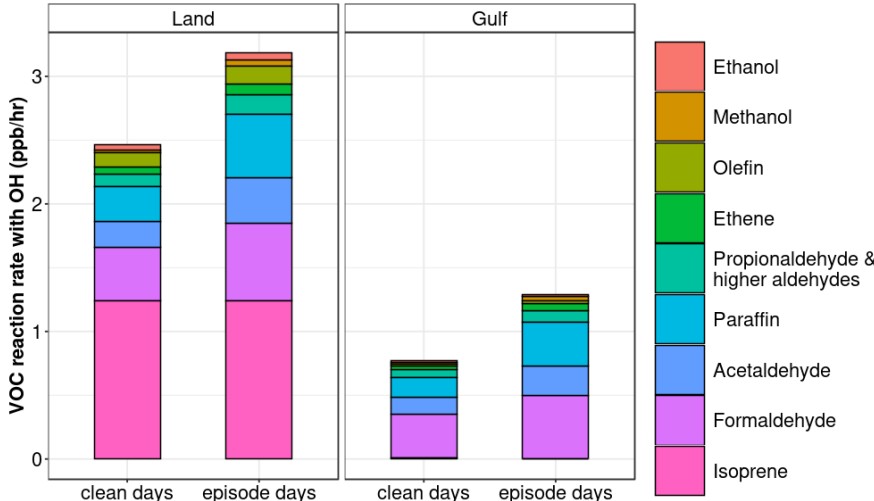

**Figure 7. OH reaction rates with different VOCs on clean days and ozone-episode days during 11:00 – 15:00 over the urban area (Land; black box in Figure 3) and the Gulf of Mexico (Gulf; black box in Figure 2).**

In summary, O₃ chemistry is the major process responsible for the high O₃ mixing ratios over the Gulf of Mexico during the study period. The VOC species with a long lifetime advected from the northeast increase over land and water, leading to a region-wide enhancement of PO₃ under a NOx-limited regime. The downwind transport of NOₓ from the Ship Channel also expands the VOC-limited area towards the west side of Galveston Bay and the Gulf of Mexico, contributing to the higher-than-normal PO₃.



**3.3 Case studies**
Although the above analysis reveals the general reasons responsible for the high offshore O$_3$ events, the multiple-
day average can miss out on some important aspects regarding the causes of these events. In this section, we selected
two case days, September 9 and October 7, to further demonstrate the development process of high O$_3$ in detail.
**3.3.1 Case study of September 9, 2021**

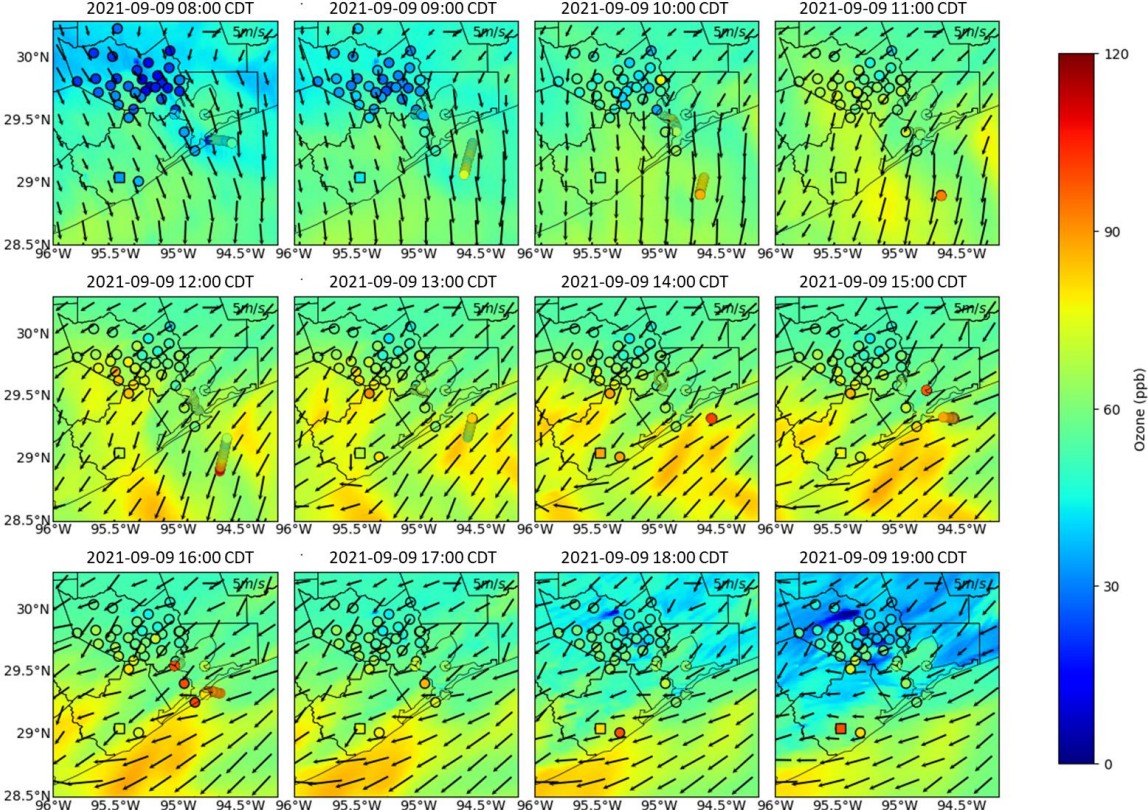


**Figure 8. Hourly simulated ozone distributions (color contours) from 08:00 to 19:00 (CDT) on September 9 overlaid with**
**winds (arrows). Onshore and offshore dots indicate ozone from CAMS sites and boat observations. The square mark**
**highlights the Lake Jackson CAMS site.**





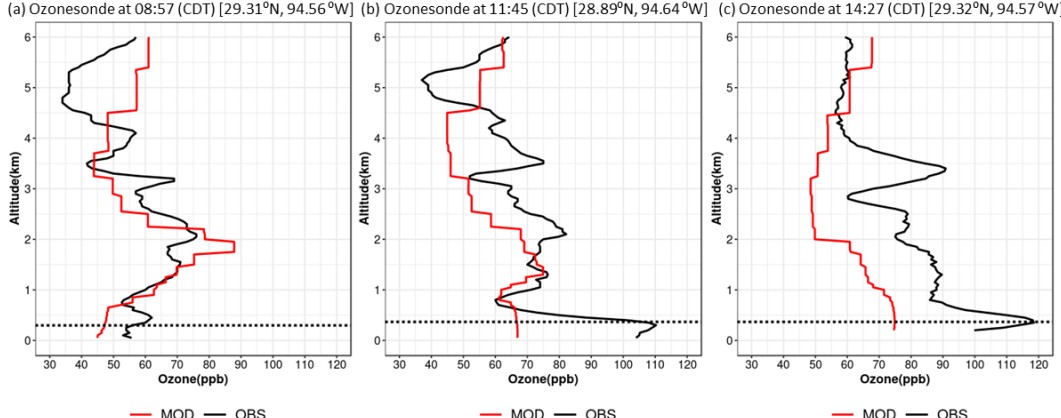

**Figure 9. Ozone vertical profiles from ozonesondes (black line) and model simulations (red line) at 08:57 (a), 11:45 (b), and 14:27 (c) on September 9. Black dash lines indicate the observed boundary layer height.**

Multiple CAMS sites exceeded the 70 ppb MDA8 $O_3$ standard on September 9, with the Red Eagle boat sampling the up to 115 ppb 1-minute $O_3$ in the Gulf of Mexico off the coast of Galveston Island. The hourly progression of the observed and simulated $O_3$ is displayed in Figure 8, overlaid with modeled winds. In the morning, the study area was dominated by northerly winds bringing the fresh emissions offshore while the pontoon boat was sampling over the west side of Galveston Bay and the Red Eagle boat was traveling in the Gulf of Mexico off the coast of Galveston Island. The ozonesonde launched near 09:00 shows a moderate level of $O_3$ (~55 ppb) below the shallow marine boundary layer of 200 m overlaid by a residual layer with a maximum $O_3$ mixing ratio of 63 ppb at ~500 m (Figure 9a). Around 11:00-12:00, with high solar radiation, the seaward-transported emissions formed $O_3$ through photochemical reactions over water, which was captured by the Red Eagle boat with an hourly peak $O_3$ mixing ratio of 92 ppb (Figure 10a). Correspondingly, the $O_3$ vertical profile from the 11:45 balloon launch at the Red Eagle deck recorded the highest $O_3$ of 110 ppb at ~315 m (Figure 9b).

However, the model missed these peak values because the simulated wind speed is up to 4 m/s higher than observations (Figure 10c), making the plume advect faster. This also leads to a two-hour earlier arrival of the modeled $O_3$ peak at the Lake Jackson coastal site (square mark in Figure 8) than the observed first peak at 14:00 (Figure 10a). At the same time, another plume was brought into the Gulf of Mexico from the east boundary of the domain as the wind directions changed from north to east. As the Red Eagle boat steered back to Galveston Island, all three boats sampled this plume at 14:00-17:00, resulting in the second $O_3$ peak at the Red Eagle boat and the only $O_3$ peak at the other two boats. The ozonesonde launched at 14:27 from the Red Eagle boat (Figure 9c) observed $O_3$ reaching 118 ppb in the plume at ~370 m. This plume was continuously transported southwestward and reached the Lake Jackson site at 19:00, producing a second $O_3$ peak. Due to the overestimated wind speed and the simulated wind direction not completely veering to the east as observations (about 100° in Figure 10b), the model failed to predict the timing and the magnitude of the $O_3$ peaks caused by the second plume. The process analysis on this day over the Gulf of Mexico (black box in Figure 2) shows ADV, in addition to CHEM, contributes to the enhanced $O_3$



levels at 10:00 and 13:00 (Figure S7), which respectively corresponds to the two plumes under northerly and
easterly winds and highlights the importance of regional transport. This also demonstrates that the contributions
from ADV to the increase of $O_3$ can be high on some specific cases, which can be averaged out in our composite
analysis of Figure 5.

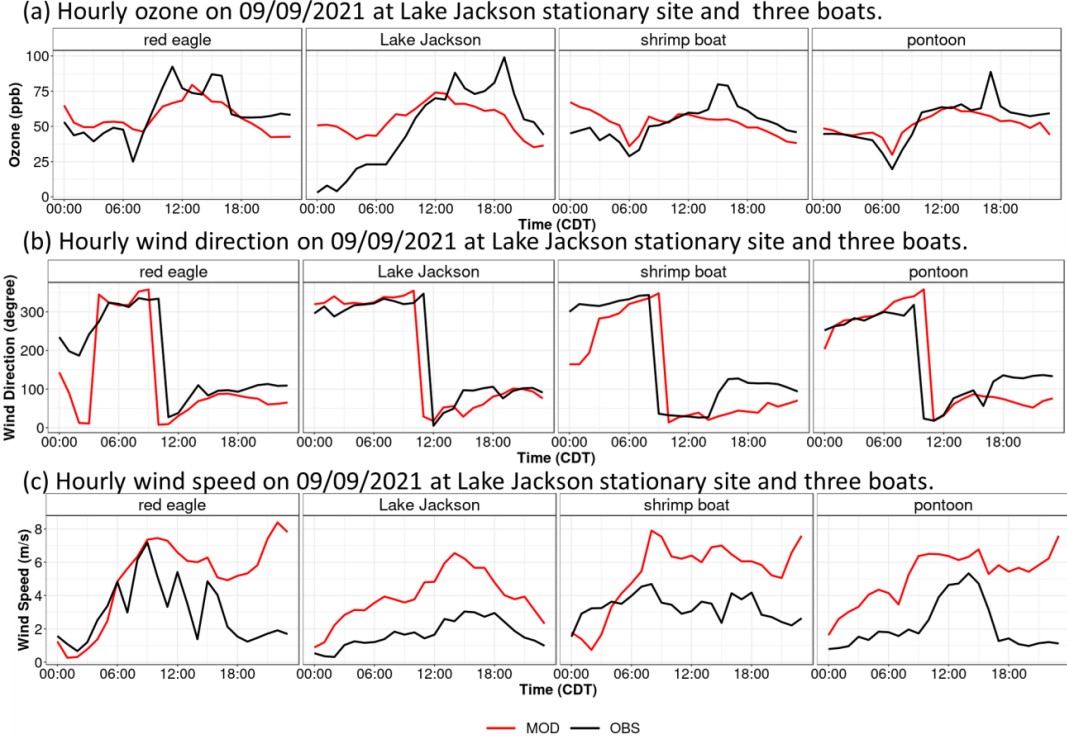


**Figure 10. Hourly ozone (a), wind direction (b), and wind speed (c) on September 9 from observations at the Lake Jackson**
**CAMS site (square mark in Figure 8) and three boats (black) in comparison with model simulations (red).**


In summary, the wind direction changes from the north to the east on September 9 caused two $O_3$ peaks, as captured
by the Red Eagle boat and the Lake Jackson site. This corresponds to the two simulated ozone plumes shown in the
maps. One plume is produced locally and the other is transported from the eastern boundary of the domain. The
model overestimates the wind speed, and the simulated wind direction does not change entirely to easterly, leading
to lower or totally missed and temporally mismatched $O_3$ peaks relative to observations.





**3.3.2 Case study of October 7, 2021**

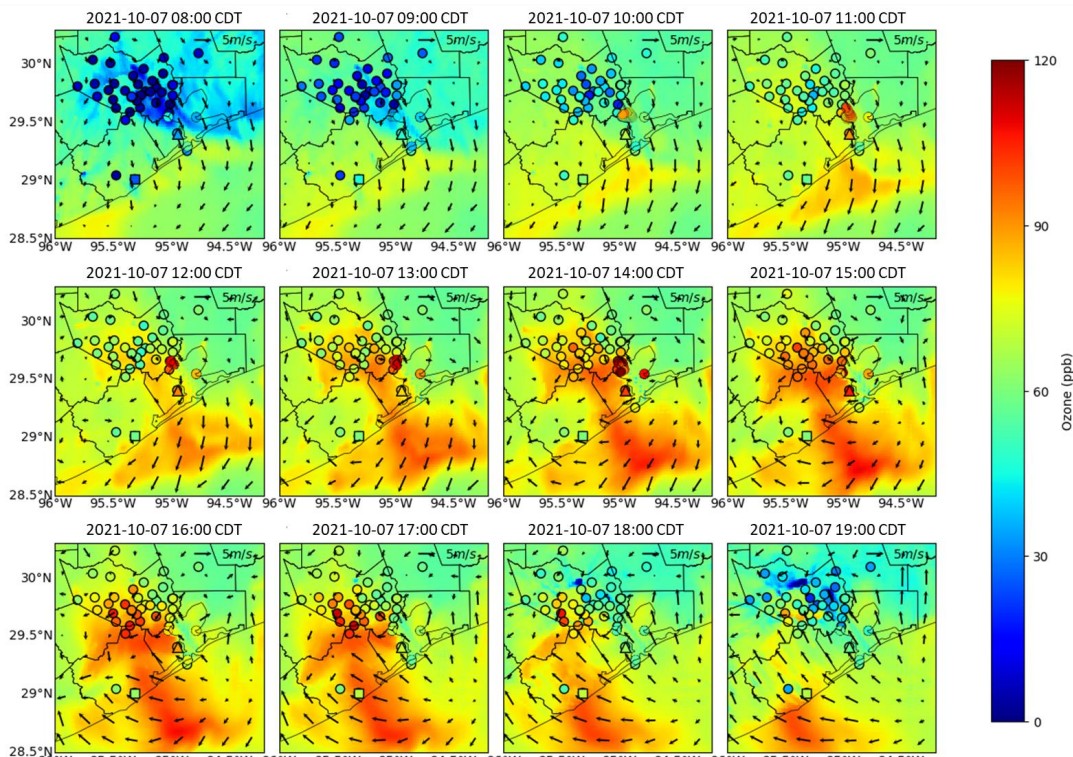

**Figure 11. Same as Figure 8 but on October 7 with the square and triangle marks representing the Oyster Creek and Texas**
**City CAMS sites, respectively.**
On October 7, the pontoon boat observed the highest one-minute $O_3$ concentration (135 ppb) throughout the entire
campaign period. This day started with weak northwesterly winds in the morning under post-frontal conditions,
leading to high $O_3$ concentrations along the Gulf coast (Figure 11). The winds transitioned to northeasterly near
11:00 (Figure 12b), marking the onset of the Galveston Bay breeze at the pontoon and shrimp boat and the Texas
City site (triangle label in Figure 11) and the Gulf breeze at the Oyster Creek site (square label in Figure 11), both
accompanied by an increase of $O_3$ (Figure 12a) and wind speed (Figure 12c). By contrast, the model predicted a late
onset of the Bay/Gulf breezes by two to three hours with a generally higher wind speed than was observed.
Afterward, the wind directions further shifted to the east to southeast between 14:00 to 18:00 as the Gulf breezes
propagated to all four locations in Figure 12b, causing the highest $O_3$ mixing ratios therein. Similarly, the model
overestimated the Gulf breeze intensity, leading to the underestimation of $O_3$ at the three locations along Galveston
Bay. The model also continuously overestimated the moderate level of $O_3$ (60-70 ppb) at the Oyster Creek site under
the Gulf breeze from 11:00 to 20:00, implying that the lifetime of $O_3$ or its precursors over water was likely
overpredicted. Different from September 9, the process analysis on this local-scale event indicates CHEM is the
major process leading to high $O_3$ concentrations over the Gulf of Mexico (Figure S8). ADV only contributes to the



increase of $O_3$ at 08:00-09:00, corresponding to the offshore transport of $O_3$ in the morning under northwesterly
winds.

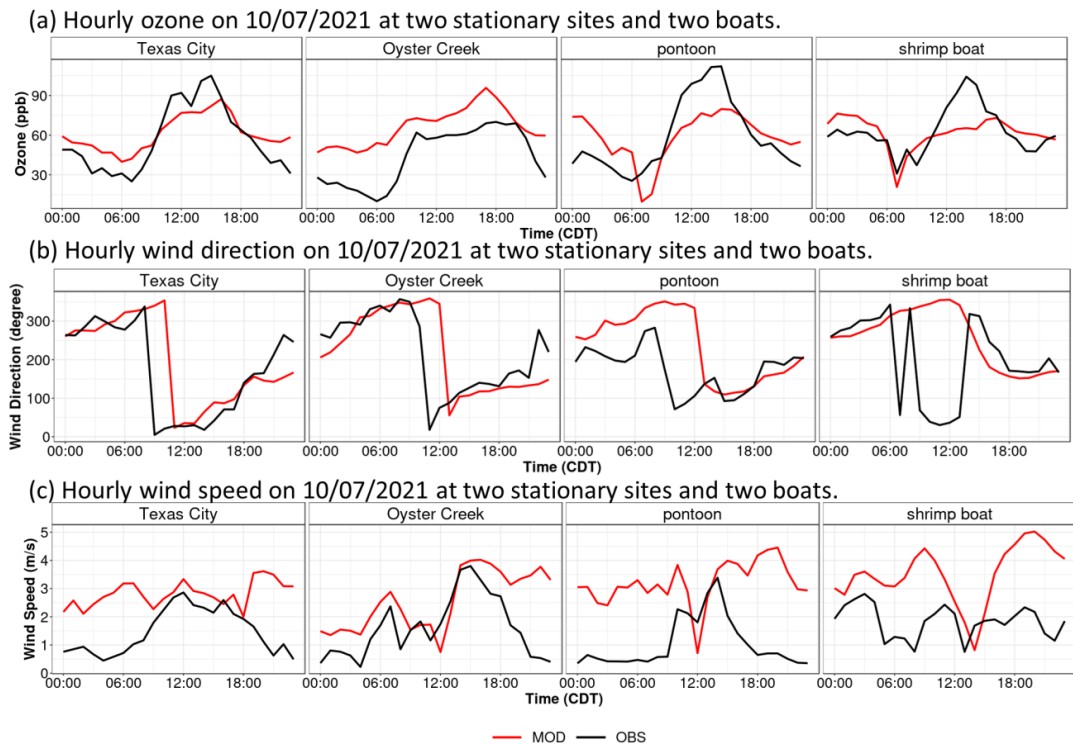


**Figure 12. Same as Figure 10 but on October 7 with the Texas City (triangle mark in Figure 11) and Oyster Creek (square
mark in Figure 11) CAMS sites and two boats.**


To sum up, the high $O_3$ event on October 7 was related to the mesoscale Galveston Bay and Gulf breeze
recirculation. Two boats and the Texas City site captured the start of the Bay breeze at ~11:00 and the development
of the Gulf breeze at 14:00 – 18:00, the latter of which leads to peak hourly $O_3$ by bringing the aged $O_3$ and
emissions back to land. Affected continuously by the Gulf breeze from 11:00 to 20:00, $O_3$ at the Oyster Creek site
stayed at 60 – 70 ppb. The model predicts the onset of the Bay and Gulf breezes two to three hours late with higher
wind speed, causing the delayed and lower $O_3$ peaks along Galveston Bay.
**4 Conclusions**
As part of the TRACER-AQ 2021 field campaign in the Houston area, three boats, a UH pontoon boat and two
commercial vessels, equipped with an automatic sampling system and ozonesonde launches were deployed in
Galveston Bay and the Gulf of Mexico from July to October. The resulting datasets, including the surface and



vertical $O_3$ concentrations and various meteorological parameters, provide a unique opportunity to evaluate the
performance of TCEQ's regulatory WRF-CAMx modeling system regarding its ability to capture the high offshore
$O_3$ events. Driven by the optimized WRF meteorological outputs, the CAMx model can satisfactorily capture the
spatiotemporal variability of daytime $O_3$ for the three boats (R > 0.70) with an overall $4 - 8$ ppb $(9\% - 22\%)$
overestimation mainly caused by the high positive biases on clean days. During high-$O_3$ events, the model tends to
underestimate $O_3$ by 5 ppb near the surface and by 10 ppb up to 4 km aloft.
The reasonable model performance provides credibility for relying on the model's process analysis tool to
investigate the factors responsible for the high-$O_3$ episodes over the Gulf of Mexico. The results show that $O_3$
chemistry is the major process leading to high $O_3$ concentrations relative to clean conditions. A region-wide increase
of long-lived VOC species through advection, such as paraffin, formaldehyde, and acetaldehyde, accelerated $O_3$
production rates under a $NO_x$-limited regime. In the meantime, the enhanced VOCs can produce more $O_3$ near
western Galveston Bay and off the Gulf coast under high-$NO_x$ concentrations brought by the northeasterly winds
from the Houston Ship Channel. Thus, the higher $O_3$ chemical production over water can be from both NOx- and
VOC-limited regimes.
Two cases, September 9 and October 7, were then selected to illustrate the development of high-$O_3$ events further.
Both cases involved north/northeast morning winds transporting the inland emissions toward the sea, shifting to the
east/southeast in the afternoon, and transporting the offshore $O_3$ and its precursors to the land. Therefore, well-
represented wind conditions are of great importance for air quality models to accurately capture the timing and
magnitude of elevated $O_3$ levels in these cases. However, the two cases differ in terms of atmospheric scale. The
event on September 9 was influenced by a large-scale circulation with regionally homogeneous wind conditions.
The easterly winds in the afternoon brought a second air plume from the eastern boundary of the domain following
the first locally produced plume, illustrating the contributions of regional advection, in addition to chemistry, to the
high $O_3$ mixing ratios in this case. Conversely, the October 7 case was dominated by the mesoscale development of
Bay and Gulf breezes, characterized by a generally lower wind speed and higher $O_3$ level. Double $O_3$ peaks can also
be observed near Galveston Bay, such as the Texas City site in this case, corresponding to the arrival of the Bay and
Gulf breezes, respectively. The model mispredicted the timing of the wind direction shift and overestimated the
wind speed in both cases, leading to the temporally mismatched and numerically buffered $O_3$ peaks.
This study reveals the important role of chemical $O_3$ production over Galveston Bay and the Gulf of Mexico from
precursors emitted from adjacent land and the Ship Channel or transported regionally from the northeastern states.
The high $O_3$ produced offshore can then be transported back to land and cause $O_3$ exceedances at the air quality
monitors. Therefore, local and regional emissions need to be stringently regulated to reduce the frequency of such
events. Additionally, wind conditions are critical meteorological factors leading to these high-$O_3$ episodes and thus
need to be well represented in photochemical models to have an accurate air quality forecast in urban coastal
regions.



**Acknowledgments**
This research was supported by the Texas Commission for Environmental Quality (TCEQ, Grant Numbers 582-22-
31544-019) and the State of Texas Air Quality Research Program (AQRP, Project 20-008). The findings, opinions,
and conclusions are the work of the author(s) and do not necessarily represent the findings, opinions, or conclusions
of the TCEQ or AQRP. We acknowledge the individuals and groups who collected and shared the TRACER-AQ
2021 filed campaign datasets.
**Data Availability**
CAMx and WRF models are publicly available at https://www.camx.com/ and
https://www2.mmm.ucar.edu/wrf/users/download/get_source.html, respectively. CAMS data can be downloaded
from the TAMIS web interface (https://www17.tceq.texas.gov/tamis/index.cfm?fuseaction=home.welcome), and
other campaign data is archived in the TRACER-AQ website (https://www-air.larc.nasa.gov/cgi-
bin/ArcView/traceraq.2021).
**Competing interests**
The authors declare that they have no conflict of interest.
**Author contributions**
YW conceived the research idea. WL, XL and ES conducted the model simulation. TG, JF and PW provided the
field observations. WL performed the data analysis and drafted the initial manuscript. All authors contributed to the
interpretation of the results and the preparation of the manuscript

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
