# Peer review of "Understanding offshore high-ozone events during TRACER-AQ 2021 in Houston: Insights from WRF-CAMx photochemical modeling"

_EGUsphere, 2023_

## Author Comment (AC1)

**Reply to Reviewers**

We sincerely appreciate all the reviewers for their constructive comments to improve the manuscript. Their comments are reproduced below followed by our responses in blue. The corresponding edits in the manuscript are highlighted with track changes.

**Reviewer #1:**

General Comments:

This study investigates the high offshore ozone events in the Houston area by combining modelling study with data from the TRACER-AQ field campaign. This is a very interesting study and the paper is in general well written. I only have a few relatively minor comments for the authors to address.

Specific Comments:

(1). Indeed, wind conditions can be very important meteorological factors affecting the high ozone events, but I wonder if it is possible to examine other meteorological conditions. For example, some studies have shown that certain meteorological conditions such as fumigation can lead to the "touch-down" of air pollutants from above and thus significantly enhance their surface concentrations. This can be quickly examined by looking at the atmospheric stability or vertical profile of temperature. Of course, I understand these temperature data may not always be available, but worthy to check.

**Response**: Thanks for the suggestion. To verify whether the fumigation process also contributes to high ozone, we investigated the vertical profiles of potential temperature measured by ozonesondes and simulated by the model in the afternoon at an inland site UH, a coastal site La Porte, and two offshore sites over Galveston Bay and the Gulf of Mexico (Figure R1). Indeed, the potential temperature at all four sites shows a positive gradient at ~1.5km on high-ozone days (solid lines in Figure R1), which indicates the location of the inversion layer. High ozone in this layer can be mixed down to the surface as the daytime boundary layer develops and penetrates the inversion layer. This phenomenon is described in detail in our another study focusing on the boundary layer structure during the September ozone episodes of the TRACER-AQ campaign (Liu et al., 2023). On the contrary, the positive gradient of potential temperature is not pronounced during clean days (dashed lines in Figure R1). The model generally captures this feature with high correlation coefficients (R>0.9). This also corresponds to the results of the process analysis in Figure 5 in which vertical diffusion is the secondary process leading to high ozone events. Figure R1 was inserted as Figure S6 with the related texts added in Lines 246-249.

[Figure]

**Figure R1**: Potential temperature (θ) vertical distribution from the afternoon (12:00-18:00) ozonesonde launches (Obs; black lines) and simulations (Mod; red lines) at UH, La Porte, and Galveston Bay averaged on clean days (dashed lines) and ozone-episode days (solid lines). The Gulf of Mexico only sampled ozone on high-ozone days.

(2). In abstract - "The region-wide increase of long-lived VOCs through advection not only leads to more O3 production under a NOx-limited regime but also …" – this sentence is confusing (if it' indeed in NOx-limited regime, then the impacts of VOC change should be quite limited?), so I suggest rewriting this part.

**Response**: The increase of VOCs will transit ozone formation to be more sensitive to $NO_x$, which suggests more ozone will be generated under the same level of $NO_x$. In addition, ozone will fall under $NO_x$-limited regime more frequently with a higher level of VOCs as shown by the frequency changes in Figure S7 (old Figure S5). We changed the texts to avoid confusion.

**Reviewer #2**

General Comments:

The manuscript has presented a modelling analysis of the high ozone events over the offshore areas of Galveston Bay and the Gulf of Mexico observed by the TRACER-AQ field campaign. Six episodes with MDA8 ozone levels above 70 ppbv were analyzed using the surface site, ship, and balloon measurements interpreted by the WRF-CAMx modelling system and its process analysis tools. The results demonstrated the important role of chemical ozone production over

the offshore areas from precursors emitted from adjacent land, and then high ozone could be transported back to land causing ozone exceedances.

The findings of this study are important for understanding the ozone variations in Houston and surrounding regions. The manuscript is overall well-organized and presented. The topic well fits the scope of ACP. I thus recommend acceptance for publication after the following minor comments have been addressed.

Specific Comments:
(1) Section 2.2, Emissions in the model
A more detailed description of the emissions may be required in this section. Besides anthropogenic emissions, does the CAMx model include any natural emissions, such as biomass burning, soil, and lightning? If not, how would these emissions affect the model results? Any ship emissions were included in this region?
**Response**: Except for anthropogenic emissions, biogenic emissions generated from the Biogenic Emission Inventory System (BEIS) are also used in the model simulation. Wildfire emissions are based on the Fire INventory from NCAR (FINNv2). There are no lighting emissions included in the model, which could potentially cause the negative bias of the ozone. Ship emissions are estimated from the Gulfwide Emissions Inventory (GWEI). These descriptions are added to Lines 132-135 of the main texts.

(2) Section 2.2, Line 113
We usually thought that simulations with nudging or with reinitializing would be better. Can you comment more here? Were there significant differences among the simulations, and with no nudging or reinitializing the results were statistically significant better?
**Response**: Detailed evaluations of different WRF configurations can be found in Liu et al. (2023). Here we reproduced the list of model experiments in Table R1 and the evaluation metrics against inland CMAS sites and offshore boat observations in Table R2-3.

**Table R1.** List of model experiments.

| Simulations | BC Meteorology | PBL | Microphysics | Nudging | Reinitializing |
|---|---|---|---|---|---|
| [Base] | NCEP FNL | MYNN | 2M | No | No |
| [WSM6] | NCEP FNL | MYNN | WSM6 | No | No |
| [YSU] | NCEP FNL | YSU | 2M | No | No |
| [ACM2] | NCEP FNL | ACM2 | 2M | No | No |
| [ERA5] | ECMWF ERA5 | MYNN | 2M | No | No |
| [HRRR] | HRRR | MYNN | 2M | No | No |
| [Nudged2] | NCEP FNL | MYNN | 2M | Yes | No |
| [Reinit] | NCEP FNL | MYNN | 2M | No | Yes |

**Table R2.** Performance metrics of spatiotemporal variability between CAMS-observed and WRF-modeled meteorology during the high-ozone episodes. Hourly meteorology at all stations is used for the calculation of performance metrics below. All metrics have the same unit as meteorological variables, except that the correlation coefficient (R) and normal mean bias (NMB) are unitless.

| Variables | Simulation | OBS | MOD | R | NMB | MB | MAE | RMSE |
|---|---|---|---|---|---|---|---|---|
| Temperature (°C) | [Base] | 26.18 | 25.82 | 0.88 | -0.01 | -0.36 | 1.69 | 2.15 |
| | [WSM6] | | 25.84 | 0.89 | -0.01 | -0.35 | 1.57 | 1.99 |
| | [YSU] | | 26.29 | 0.89 | 0.00 | 0.11 | 1.65 | 2.11 |
| | [ACM2] | | 25.95 | 0.86 | -0.01 | -0.23 | 1.76 | 2.23 |
| | [ERA5] | | 24.91 | 0.85 | -0.05 | -1.28 | 2.17 | 2.71 |
| | [HRRR] | | 26.12 | 0.89 | 0.00 | -0.06 | 1.59 | 2.05 |
| | [Nudged] | | 25.92 | 0.92 | -0.01 | -0.26 | 1.43 | 1.84 |
| | [Re-init] | | 25.69 | 0.92 | -0.02 | -0.49 | 1.41 | 1.77 |
| Relative humidity (%) | [Base] | 60.12 | 60.94 | 0.76 | 0.01 | 0.82 | 10.25 | 13.04 |
| | [WSM6] | | 62.21 | 0.78 | 0.03 | 2.09 | 9.85 | 12.28 |
| | [YSU] | | 58.45 | 0.80 | -0.03 | -1.68 | 9.54 | 12.31 |
| | [ACM2] | | 62.73 | 0.71 | 0.04 | 2.60 | 11.40 | 14.71 |
| | [ERA5] | | 64.21 | 0.77 | 0.07 | 4.08 | 10.55 | 12.76 |
| | [HRRR] | | 57.82 | 0.79 | -0.04 | -2.30 | 9.13 | 12.13 |
| | [Nudged] | | 64.63 | 0.82 | 0.08 | 4.51 | 9.54 | 12.05 |
| | [Re-init] | | 62.57 | 0.84 | 0.04 | 2.45 | 8.37 | 10.66 |
| Wind speed (m/s) | [Base] | 0.67 | 1.29 | 0.35 | 0.59 | 1.01 | 1.40 | 1.70 |
| | [WSM6] | | 1.67 | 0.37 | 0.61 | 1.04 | 1.39 | 1.72 |
| | [YSU] | | 0.80 | 0.39 | 0.75 | 1.29 | 1.55 | 1.87 |
| | [ACM2] | | 1.16 | 0.38 | 0.66 | 1.12 | 1.44 | 1.77 |
| | [ERA5] | | 1.76 | 0.43 | 0.64 | 1.09 | 1.38 | 1.66 |
| | [HRRR] | | 1.00 | 0.54 | 0.49 | 0.83 | 1.12 | 1.36 |
| | [Nudged] | | 0.89 | 0.55 | 0.30 | 0.51 | 0.96 | 1.20 |
| | [Re-init] | | 1.14 | 0.61 | 0.48 | 0.82 | 1.07 | 1.31 |
| Wind direction (deg) | [Base] | 87.76 | 72.32 | 0.43 | -0.05 | -7.67 | 56.5 | 73.36 |
| | [WSM6] | | 72.56 | 0.38 | -0.04 | -5.51 | 56.41 | 72.93 |
| | [YSU] | | 53.26 | 0.41 | -0.08 | -12.14 | 60.30 | 77.29 |
| | [ACM2] | | 54.87 | 0.37 | -0.07 | -10.64 | 64.15 | 81.29 |
| | [ERA5] | | 47.32 | 0.43 | -0.07 | -10.92 | 58.05 | 74.83 |
| | [HRRR] | | 92.51 | 0.61 | -0.02 | -3.43 | 40.16 | 57.55 |
| | [Nudged] | | 93.29 | 0.48 | 0.02 | 3.00 | 46.05 | 64.70 |
| | [Re-init] | | 109.03 | 0.47 | 0.00 | -0.32 | 39.99 | 57.67 |

**Table R3.** Performance metrics of spatiotemporal variability between boat-observed and WRF-modeled meteorology during the high-ozone episodes. 1-minute meteorology is used for the calculation of performance metrics below. All metrics have the same unit as meteorological variables, except that the correlation coefficient (R) and normal mean bias (NMB) are unitless.

| Variables | Simulation | OBS | MOD | R | NMB | MB | MAE | RMSE |
|---|---|---|---|---|---|---|---|---|
| Temperature (°C) | [Base] | 26.55 | 26.45 | 0.77 | 0.00 | -0.11 | 1.71 | 2.14 |
| | [WSM6] | | 26.50 | 0.75 | 0.00 | -0.05 | 1.77 | 2.20 |
| | [YSU] | | 26.78 | 0.78 | 0.01 | 0.22 | 1.71 | 2.10 |
| | [ACM2] | | 26.51 | 0.75 | 0.00 | -0.04 | 1.78 | 2.21 |
| | [ERA5] | | 24.85 | 0.75 | -0.06 | -1.70 | 2.21 | 3.00 |
| | [HRRR] | | 26.30 | 0.75 | -0.01 | -0.25 | 1.89 | 2.29 |
| | [Nudged] | | 26.30 | 0.87 | -0.01 | -0.25 | 1.26 | 1.65 |
| | [Re-init] | | 26.53 | 0.76 | 0.00 | -0.02 | 1.71 | 2.15 |
| Relative humidity (%) | [Base] | 60.96 | 70.24 | 0.64 | 0.15 | 9.28 | 11.95 | 14.59 |
| | [WSM6] | | 71.09 | 0.61 | 0.17 | 10.14 | 11.76 | 14.38 |
| | [YSU] | | 68.20 | 0.65 | 0.12 | 7.24 | 10.96 | 13.29 |
| | [ACM2] | | 69.35 | 0.56 | 0.14 | 8.40 | 12.75 | 15.33 |
| | [ERA5] | | 74.38 | 0.60 | 0.22 | 13.42 | 14.66 | 17.23 |
| | [HRRR] | | 69.20 | 0.70 | 0.14 | 8.24 | 10.38 | 12.68 |
| | [Nudged] | | 73.35 | 0.75 | 0.20 | 12.39 | 12.87 | 14.92 |
| | [Re-init] | | 69.68 | 0.67 | 0.14 | 8.72 | 10.25 | 12.42 |
| Wind speed (m/s) | [Base] | 0.73 | 2.47 | 0.16 | 0.74 | 1.67 | 2.20 | 2.78 |
| | [WSM6] | | 2.62 | 0.14 | 0.82 | 1.85 | 2.33 | 2.92 |
| | [YSU] | | 2.17 | 0.13 | 0.99 | 2.22 | 2.63 | 3.19 |
| | [ACM2] | | 1.99 | 0.15 | 0.92 | 2.07 | 2.49 | 3.09 |
| | [ERA5] | | 1.89 | 0.22 | 0.78 | 1.74 | 2.21 | 2.72 |
| | [HRRR] | | 1.68 | 0.52 | 0.59 | 1.32 | 1.69 | 2.05 |
| | [Nudged] | | 1.75 | 0.37 | 0.41 | 0.92 | 1.57 | 1.96 |
| | [Re-init] | | 2.02 | 0.30 | 0.69 | 1.55 | 2.00 | 2.41 |
| Wind direction (deg) | [Base] | 144.15 | 118.78 | 0.32 | -0.08 | -11.45 | 57.74 | 75.38 |
| | [WSM6] | | 113.5 | 0.26 | -0.13 | -19.10 | 60.40 | 77.29 |
| | [YSU] | | 135.77 | 0.26 | -0.11 | -16.44 | 63.52 | 81.13 |
| | [ACM2] | | 125.25 | 0.27 | -0.11 | -17.20 | 68.93 | 85.92 |
| | [ERA5] | | 96.69 | 0.18 | -0.17 | -25.20 | 69.00 | 85.30 |
| | [HRRR] | | 137.93 | 0.58 | -0.08 | -12.53 | 41.54 | 58.16 |
| | [Nudged] | | 146.95 | 0.45 | -0.05 | -7.68 | 47.87 | 65.51 |
| | [Re-init] | | 146.96 | 0.62 | -0.10 | -14.98 | 42.98 | 59.66 |
| | [Base] | 855.58 | 499.27 | 0.32 | -0.42 | -356.30 | 529.63 | 699.67 |

| Boundary layer height (m) | [WSM6] | 526.69 | 0.30 | -0.38 | -328.88 | 526.38 | 691.82 |
| | [YSU] | 322.22 | 0.30 | -0.62 | -533.36 | 612.29 | 817.16 |
| | [ACM2] | 443.60 | 0.30 | -0.48 | -411.97 | 562.12 | 747.06 |
| | [ERA5] | 464.75 | 0.47 | -0.46 | -390.83 | 507.51 | 680.30 |
| | [HRRR] | 671.27 | 0.38 | -0.22 | -184.31 | 461.30 | 637.68 |
| | [Nudged] | 462.09 | 0.41 | -0.46 | -393.48 | 516.18 | 696.37 |
| | [Re-init] | 569.57 | 0.25 | -0.33 | -286.00 | 518.21 | 689.22 |

The WRF model generally reproduces observed temporal variability and spatial distribution in key meteorological parameters with most of the correlation coefficients higher than 0.5. However, the model, regardless of configuration settings, shows persistent low biases in PBL heights, low biases in air temperatures, high biases in relative humidity, and high biases in wind speed. While [HRRR], [Nudged], and [Reinit] configurations stand out as the best simulations based on campaign-wide statistics, the performance of these three is indistinguishable. No one is significantly and consistently better than the others. Considering that [Nudged] requires additional observational datasets and [Reinit] needs to automate the model running process, [HRRR] is the easiest and the most effective option to reproduce meteorology for computationally expensive chemistry simulations.

(3) Section 2.2, Line 141

It is not clear from Figure S1 that the redistributed emissions performed better. Can you please provide any comparison metrics to show this feature?
**Response**: As suggested, we summarized the performance metrics of the two simulations using redistributed emissions and the Flexi-nesting option in Table R4. All the metrics are slightly improved by the redistributed emissions, except for R staying at the same value of 0.87. Since we mainly conducted special treatments to on-road emissions, bigger changes are found along the major highways as shown in Figure S1c. The redistributed emissions are expected to better capture the ozone distribution on the main roads. Table R4 was inserted in the supplement file as Table S1.

**Table R4.** Performance metrics of regional-mean hourly $O_3$ between CAMS observations and simulations. All metrics have the same unit as their corresponding variables, except that the correlation coefficient (R) and normal mean bias (NMB) are unitless.

| Variables | Simulation | OBS | MOD | R | NMB | MB | MAE | RMSE |
|---|---|---|---|---|---|---|---|---|
| $O_3$ (ppb) | Flexi-nesting | 23.71 | 29.01 | 0.87 | 22.38 | 5.31 | 7.41 | 8.59 |
| | Redistribution | | 28.90 | 0.87 | 21.90 | 5.19 | 7.34 | 8.54 |

(4) Section 3.1, Model evaluation with ozonesonde

The large model underestimates relative to ozonesonde measurements for the episode days may need some more discussion. The model underestimates appear to be persistent throughout the low troposphere up to 5 km as shown in Figure 4. It is not clear whether the model missed any other processes in addition to the wind factors. Please provide some further clarification.

**Response**: All the ozonesondes available for generating Figure 4 are from the two high-ozone episodes of September 6-11 and 23-26. Two episodes are featured by a high-ozone plume at the layer near 3 km observed by the ozone lidar at La Porte as shown by Figure 8 in our another study (Liu et al., 2023), which is adapted as Figure R2 below. The plumes are also captured by ozonesondes as shown by the two example days (September 9[th] and 24[th]) of each episode over Galveston Bay (Figure R3). The model failed to simulate such high-ozone plumes, indicating the long-range transport of ozone is underestimated in the model. High ozone in the plumes can be mixed down to the surface through the fumigation process as pointed out by Review #1, which further leads to the underestimation of ozone within the boundary layer. We added Figure R3 to the supplement file as Figure S3 with related texts inserted in Lines 215-219 to further explain the negative bias shown in Figure 4.

[Figure]

**Figure R2**: Time series of the vertical ozone profile from the Tropospheric Ozone lidar (TROPOZ) during two high-ozone episodes at La Porte. Black lines in each subplot represent the observed boundary layer height.

[Figure]

**Figure R3**: Ozone vertical distributions observed by ozonesondes (black) and simulated by the model (red) over Galveston Bay on September 9[th] and 24[th].

(5) Section 3.2, Page 10, Line 238

It is unclear why you examined the reactions of HO2 self-reaction and OH reaction with NO2. How were their rates linked with the production of ozone? A more detailed explanation in terms of chemistry is needed here.

**Response**: The abundance and reactivity of ozone precursors determine the ozone production regime, which can be indicated by the loss of $HO_x$ radicals ($HO_x$=OH+$HO_2$) as the termination of ozone chain reactions. Under low $NO_x$ conditions, the most important $HO_x$ loss is the self-reaction of hydroperoxyl radical ($HO_2$), producing hydrogen peroxide ($H_2O_2$), which is used to represent $NO_x$-limited ozone production. In urban areas with high $NO_x$ concentrations, the dominant sink for $HO_x$ radicals is the oxidation of $NO_2$ by OH, resulting in the production of

nitric acid ($HNO_3$). Therefore, $HNO_3$ is used to represent ozone production under a VOC-limited regime. The model uses the ratio of $P(H_2O_2)$ to $P(HNO_3)$ to determine the ozone formation mechanism. Thus, we examined the reactions of $HO_2$ self-reaction and OH reaction with $NO_2$ in order to analyze the transition of ozone chemistry during high-ozone periods. We add more explanations to Lines 154-160 when describing the modeling method.

(6) Caption of Figure 8, What is CDT?
**Response**: It means Central Daylight Time. We have added the full name in the caption.

**References:**

Liu, X., Wang, Y., Wasti, S., Li, W., Soleimanian, E., Flynn, J., Griggs, T., Alvarez, S., Sullivan, J. T., Roots, M., Twigg, L., Gronoff, G., Berkoff, T., Walter, P., Estes, M., Hair, J. W., Shingler, T., Scarino, A. J., Fenn, M., and Judd, L.: Evaluating WRF-GC v2.0 predictions of boundary layer and vertical ozone profiles during the 2021 TRACER-AQ campaign in Houston, Texas, EGUsphere, 1–33, https://doi.org/10.5194/egusphere-2023-892, 2023.